# Network-based prediction of drug combinations

Feixiong Cheng[1,2,3,4,5], István A. Kovács[1,2] & Albert-László Barabási[1,2,6,7]

Drug combinations, offering increased therapeutic efficacy and reduced toxicity, play an important role in treating multiple complex diseases. Yet, our ability to identify and validate effective combinations is limited by a combinatorial explosion, driven by both the large number of drug pairs as well as dosage combinations. Here we propose a network-based methodology to identify clinically efficacious drug combinations for specific diseases. By quantifying the network-based relationship between drug targets and disease proteins in the human protein–protein interactome, we show the existence of six distinct classes of drug–drug–disease combinations. Relying on approved drug combinations for hypertension and cancer, we find that only one of the six classes correlates with therapeutic effects: if the targets of the drugs both hit disease module, but target separate neighborhoods. This finding allows us to identify and validate antihypertensive combinations, offering a generic, powerful network methodology to identify efficacious combination therapies in drug development.

[1] Center for Complex Networks Research and Department of Physics, Northeastern University, Boston, MA 02115, USA. [2] Center for Cancer Systems Biology and Department of Cancer Biology, Dana-Farber Cancer Institute, Boston, MA 02215, USA. [3] Genomic Medicine Institute, Lerner Research Institute, Cleveland Clinic, Cleveland, OH 44106, USA. [4] Department of Molecular Medicine, Cleveland Clinic Lerner College of Medicine, Case Western Reserve University, Cleveland, OH 44195, USA. [5] Case Comprehensive Cancer Center, Case Western Reserve University School of Medicine, Cleveland, OH 44106, USA. [6] Channing Division of Network Medicine, Department of Medicine, Brigham and Women's Hospital, Harvard Medical School, Boston, MA 02215, USA. [7] Center for Network Science, Central European University, Budapest 1051, Hungary. These authors contributed equally: Feixiong Cheng, István A. Kovács. Correspondence and requests for materials should be addressed to A.-L.B. (email: alb@neu.edu)

Combination therapy, the use of multiple drugs to improve clinical outcomes, has multiple advantages compared to monotherapy[1,2]: it offers higher efficacies or, through lower individual dosage, it can reduce the risk of adverse effects[3]. Consequently, combination therapies are widely used in the treatment of multiple complex diseases, from hypertension [4] to cancer[5] and infectious diseases[6,7]. However, the systematic identification of drug combinations that simultaneously offer high clinical efficacy and low toxicity is often driven by intuition and experience rather than established principles. There is a pressing need, therefore, for novel methodologies to facilitate the discovery of multicomponent therapy.

One approach is the systematic high-throughput testing of pairwise drug combinations, which, however, faces a combinatorial challenge: for 1000 U.S. Food and Drug Administration (FDA)-approved drugs, there are 499,500 possible pairwise combinations that should be tested over approximately 3000 human diseases and multiple dosage combinations[5,6,8,9]. We are, therefore, far from even a cursory exploration of the vast number of possible combinations with potentially positive clinical outcomes. To be sure, several machine learning-based "black-box" models have been developed to predict drug combinations[10–12], offering a modest increase in accuracy[10] over random guesses[11]. We lack, however, predictive, mechanism-driven, network-medicine-based approaches to predict efficacious drug combinations.

Network-based approaches have already offered a promising framework to identify novel insights to accelerate drug discovery[13], helping us quantify both disease–disease[14] and drug–disease[15,16] relationships. These methodological advances have raised the possibility of moving beyond the "one-drug, one-target" paradigm and exploring the "multiple-drugs, multiple-targets" possibilities offered by aiming at simultaneously modulating multiple disease proteins within the same disease module, while minimizing toxicity profiles[17–21]. In this study, we quantify the relationship between drug targets and disease proteins in the human protein–protein interactome, leading to a rational, network-based drug combination design strategy.

## Results

**Network-based proximity measure of drug–drug relationships.** Disease proteins are not scattered randomly in the interactome, but tend to form localized neighborhoods, known as disease modules[14]. An efficient way to capture network proximity between a drug $(X)$ and a disease $(Y)$ is by the $z$-score ($z = \frac{d-\mu}{\sigma}$), which relies on the shortest path lengths $d(x, y)$ between drug targets $(x)$ and disease proteins $(y)$.

$$d(X, Y) = \frac{1}{\|Y\|} \sum_{y \in Y} \min_{x \in X} d(x, y) \qquad (1)$$

The $z$-score is obtained by comparing the observed distance to a reference distance distribution between a randomly selected group of proteins of matching size and degree distribution as the disease proteins and drug targets in the human interactome. The $z$-score is applicable when the reference distribution is well described by a Gaussian (Supplementary Note 1) and was successfully used to identify well-known and new drug–disease relationships in the human interactome for monotherapy[15]. Here, we hypothesize that exploring the network-based relationship between two drugs and their targets, and the disease proteins in the disease module would help clarify the mechanism-of-action of effective drug combinations while minimizing adverse effects (Supplementary Fig. 1).

To test our hypothesis, we assembled 243,603 experimentally confirmed protein–protein interactions (PPIs) connecting 16,677

unique proteins from five data sources (Supplementary Note 2 and Supplementary Data 1). We also compiled 1978 FDA-approved or clinically investigational drugs that have at least two experimentally reported targets by pooling the high-quality drug–target binding affinity profiles from six data sources (Supplementary Data 2). In order to fully characterize the mutual relationship of two drugs and a disease module, we need a network-based proximity measure between the two drugs' targets, as well. Although it has not been tested for this purpose, in principle, we could rely on the $z$-score for this task. However, in contrast to the relatively large disease modules, each drug has only a small number of experimentally reported targets (on average 3, Supplementary Fig. 2). Therefore, the randomization procedure is not producing a Gaussian distribution as described in our previous study[15], limiting the applicability of the $z$-score. Indeed, we find that the $z$-score cannot discriminate FDA-approved pairwise combinations or clinically reported adverse drug interactions from random drug pairs (Supplementary Fig. 3). Instead of relying on randomization, therefore, we measure the network proximity of drug–target modules A, B as reflected in their target localizations using the recently introduced separation measure[14]:

$$s_{AB} \equiv \langle d_{AB} \rangle - \frac{\langle d_{AA} \rangle + \langle d_{BB} \rangle}{2} \qquad (2)$$

which compares the mean shortest distance within the interactome between the targets of each drug, $\langle d_{AA} \rangle$ and $\langle d_{BB} \rangle$, to the mean shortest distance $\langle d_{AB} \rangle$ between A–B target pairs (Fig. 1a). In $\langle d_{AB} \rangle$, targets associated with both drugs A and B have a zero distance by definition. For $s_{AB} < 0$, the targets of the two drugs are located in the same network neighborhood (Fig. 1b), while for $s_{AB} \geq 0$, the two drug targets are topologically separated (Fig. 1c). For example, imatinib (I) is an FDA-approved agent for the treatment of chronic myeloid leukemia[22]. Figure 1a, b shows that imatinib's targets are in the same network neighborhood as the targets of tandutinib (T), a FMS-like tyrosine kinase 3 (FLT3) inhibitor under phase III trial for treating acute myeloid leukemia[23]; consequently the separation score between their targets is negative, $s_{IT} = -0.35$. However, the targets of natalizumab (N), an FDA-approved monoclonal antibody for treating multiple sclerosis[24], are in a topologically distinct neighborhood from the targets of imatinib and tandutinib, having a positive $s_{IN} = 0.59$ and $s_{TN} = 0.49$, respectively (Fig. 1c).

For a network-based approach to drug combinations to be effective, we need to establish if the topological relationship between two drug–target modules, as captured by $s_{AB}$, reflects biological and pharmacological relationships, as well. We find that the network proximity ($s_{AB}$; Eq. (2)) of the targets of drug–drug pairs in the human interactome correlates with chemical, biological, functional, and clinical similarities (Fig. 1d–j), outperforming the target-overlap approaches (Supplementary Fig. 4). This suggests that each drug–target module has a well-defined network-based footprint. If the footprints of two drug–target modules are topologically separated ($s_{AB} \geq 0$), then the drugs are pharmacologically distinct. If the footprints of two drug–target modules overlap ($s_{AB} < 0$), the magnitude of the overlap is indicative of their pharmacological relationship: closer network proximity of targets of a drug pair indicates higher similarities in their chemical, biological, functional, and clinical profiles (Supplementary Fig. 5). In contrast, we find that the $z$-score measure of the targets of drug–drug pairs in the human interactome does not correlate with chemical, biological, functional, and clinical similarities (Supplementary Fig. 6). We also compared network proximity ($s_{AB}$) against five other network-based measures between targets of drug–drug pairs (Supplementary Fig. 7): (1) separation (Eq. (2)), (2) closest, (3) shortest, (4) kernel, and (5) centre (see Methods). Relying on 681 FDA-approved or

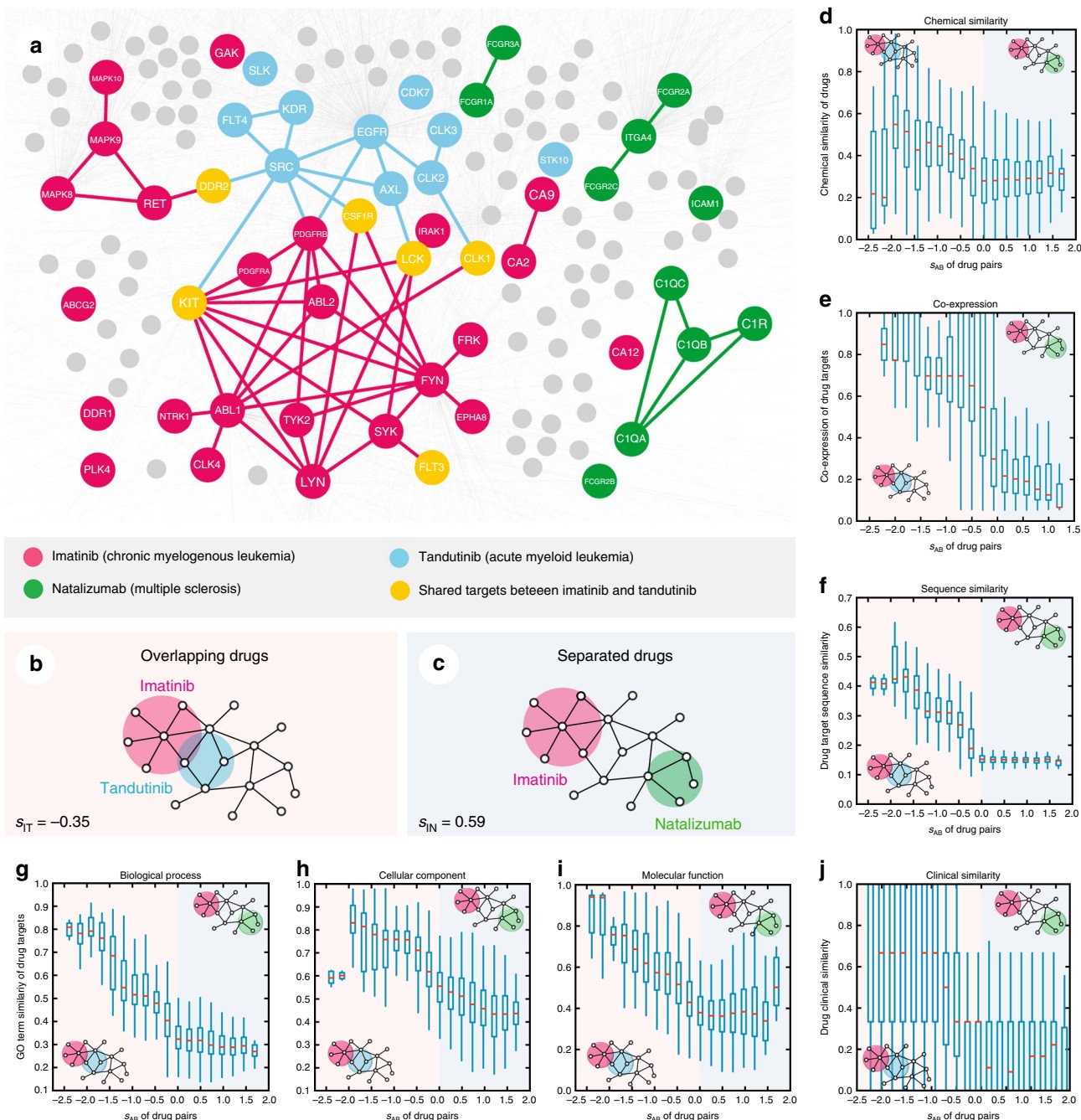

**Fig. 1** Network-based model of drug–drug relationship. **a** A subnetwork of the human interactome illustrating the network-based relationship between drug targets associated with three drugs (imatinib [I], tandutinib [T], and natalizumab [N]). **b**, **c** The definition of drug pairs that are topologically overlapping ($s_{AB} < 0$, **b**) or topologically separated ($s_{AB} \geq 0$, **c**). **d–j** The interplay between topological separation of drug pairs and five types of drug profiles: drug–drug chemical similarity (**d**); drug target-encoding gene co-expression pattern across human tissues (**e**); drug target protein sequence similarity (**f**). Using the Gene Ontology (GO) annotations, we determine for each drug how similar its associated target-encoding genes are in terms of their biological processes (**g**), cellular component (**h**), and molecular function (**i**); and clinical similarity (**j**) of drug pairs derived from Anatomical Therapeutic Chemical Classification Systems codes (see Methods). Overlapping drug pairs are highlighted in orange ($s_{AB} < 0$); topologically separated drug pairs are highlighted in blue ($s_{AB} \geq 0$)

experimentally validated pairwise drug combinations across all human diseases (Supplementary Data 3), we find that FDA-approved drug combinations have lower $s_{AB}$ compared to random drug pairs and the separation-based measure outperforms all four alternative measures as well as traditional chemoinformatics and bioinformatics approaches (Supplementary Figs. 7–9), confirming that $s_{AB}$ offers a reliable measure of drug–drug relationships within the human interactome.

**Network configurations of drug–drug–disease combinations**. Next, we turn to quantify the network-based relationship between two drug–target modules and a disease module (drug–drug–disease combinations). We find that from a network perspective, all possible drug–drug–disease combinations can be classified into six topologically distinct classes: (a) Overlapping Exposure: Two overlapping drug–target modules that also overlap with the disease module of interest (P1 in Fig. 2a); (b) Complementary Exposure:

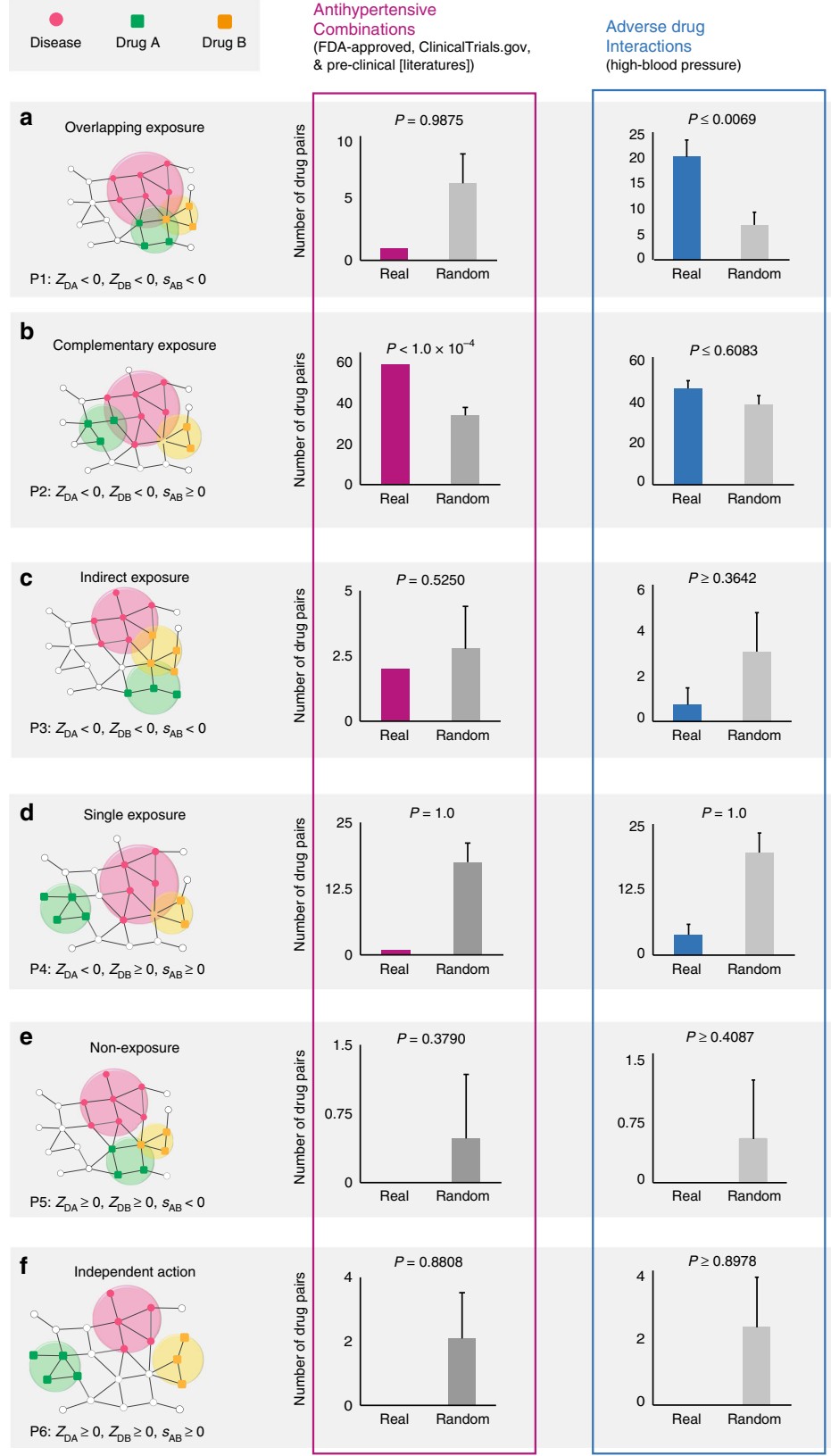

Two separated drug–target modules that overlap individually with the disease module (P2 in Fig. 2b); (c) Indirect Exposure: One drug–target module of two overlapping drug–target modules overlaps with the disease module (P3 in Fig. 2c); (d) Single Exposure: One drug–target module separated from another drug–target module overlaps with the disease module (P4 in Fig. 2d); (e) Non-exposure: Two overlapping drug–target modules are topologically separated from the disease module (P5 in Fig. 2e); and (f) Independent Action: Each of the drug–target modules and the disease module are topologically separated (P6 in Fig. 2f). The question is,

**Fig. 2** The efficacy of hypertensive drug–drug interactions. **a–f** Schematic diagrams of the six distinct classes capturing the network-based relationship between two drug–target modules and one disease module on a drug–drug–disease combination. For $s_{AB} < 0$, the two sets of drug targets overlap topologically (Fig. 1b); while for $s_{AB} \geq 0$, the two sets of drug targets are separated topologically (Fig. 1c). The z-scores ($z$), measuring the drug–disease separation, are calculated for quantifying the significance of the shortest paths between drug targets and disease proteins in the human protein–protein interactome. For $z < 0$, the drug–target module and the disease module overlap; while for $z \geq 0$, the drug–target module and the disease module are separated. Color histograms (Real) show the antihypertensive combinations (purple) and clinically reported adverse drug interactions on high blood pressure (blue), respectively. We assembled the antihypertensive combinations from three types of experimental evidences: (i) FDA-approved evidence, (ii) clinical data from Clinicaltrials.gov database, and (iii) preclinical studies from literature (Supplementary Data 3). We randomly selected the same number of adverse drug–drug interactions related to high blood pressure from 1512 clinically reported adverse drug–drug interactions (Supplementary Data 4) corresponding to the number of antihypertensive combinations using a bootstrapping algorithm in R software and this process was repeated 100 times (Supplementary Note 5). Gray boxes (Random) show random expectation. Error bars indicate the standard deviation. The P-value ($P$) is calculated by testing 10,000 permutations (Supplementary Note 5). The network-based relationships between two drug–target modules and one disease module for FDA-approved hypertensive drug combinations only are illustrated in Supplementary Fig. 11

do these six classes manifest in detectable differences in clinical efficacy for drug combinations?

To understand which of these drug–drug–disease configurations have the greatest clinical efficacy, we focus on hypertension and cancer, two diseases with a large number of FDA-approved pairwise combinations (Supplementary Data 3). We find that four out of six drug–drug–disease configurations (P3–P6 in Fig. 2c–f) do not show a statistically significant tendency to co-treat cancer (Supplementary Fig. 10c–10f) or hypertension (Fig. 2c–f and Supplementary Fig. 11c–11f). In other words, if at least one drug in a combination fails to be localized to the vicinity of the disease module, the combination does not have a therapeutic effect greater than monotherapy. This leads to our first major finding: for a drug pair to have a therapeutic effect, both drug–target modules must overlap with the disease module. This finding highlights the need to inspect the network-relationship between drug targets and disease proteins as we search for therapeutically beneficial combinations.

The second finding is that Overlapping Exposure, i.e., when the drug–target modules overlap with each other as well as with the disease module, has no statistically significant efficacy in treating the disease over monotherapy. Overlapping Exposure does, however, have statistically significant adverse effects, such as causing high blood pressure (Supplementary Data 4) compared to the control group ($P < 1.0 \times 10^{-4}$, permutation test, Overlapping Exposure [P1], Fig. 2a). For example, two FDA-approved hypertensive drugs, nadroparin and spironolactone, fall into the Overlapping Exposure (P1, Table 1) category with the hypertension disease module, in line with the observation that nadroparin increased the hyperkalemic effect (adverse effect) of spironolactone[25]. Cancer is a chronic disease with a strong genetic contribution, rarely caused by adverse drug–drug interactions. We therefore limit our testing of Overlapping Exposure in three cardiovascular outcomes, finding that Overlapping Exposure has statistically significant adverse effects on three cardiovascular outcomes (Supplementary Fig. 12): arrhythmia ($P < 1.0 \times 10^{-4}$, permutation test), heart failure ($P < 1.0 \times 10^{-4}$, permutation test), and myocardial infarction ($P < 1.0 \times 10^{-4}$, permutation test), consistent with previously reported overlapping toxicities[26].

The third key finding is that only drug pairs that have a Complementary Exposure relationship to the disease module (Fig. 2b and Supplementary Fig. 10b) show a statistically significant efficacy for drug combination therapies. Consider, for example, amiloride and hydrochlorothiazide, in the Complementary Exposure (P2 in Table 1) class. This combination has been shown to prevent glucose intolerance and to improve blood pressure control compared with monotherapy with either drug in the PATHWAY-3 trial[27]. In other words, we find that only separated drugs that individually overlap with the disease module show statistical significance in co-treatment of hypertension

($P < 1.0 \times 10^{-4}$, permutation test, Complementary Exposure in Fig. 2b and Supplementary Fig. 11b) and similarly for cancer ($P < 1.0 \times 10^{-4}$, permutation test, Supplementary Fig. 10b). Specifically, we assembled hypertensive drug combination data from FDA-approved evidence, clinical trials from the Clinicaltrials.gov database, and pre-clinical studies from literature curation (Supplementary Data 3), finding again that drug pairs with Complementary Exposure (P2) significantly tend to co-treat hypertension efficiently (Fig. 2b) compared to FDA-approved antihypertensive combinations (Supplementary Fig. 11b), indicating low selection bias. The unique network-based relationship that drug pairs with Complementary Exposure to the disease module tend to be effective drug combinations is consistent with the non-overlapping pharmacological principle in rational drug combination design[26]. In summary, characterizing the network-based relationship between drug–target modules and the disease module (Complementary Exposure) within the human interactome offers a powerful, network-based strategy for rational drug combination design.

**Network-based uncovering of hypertensive drug combinations.** The finding that Complementary Exposure is predictive of effective drug combinations prompts us to offer network-based predictions of new drug combinations using hypertension data as a validation set (Supplementary Data 3). Such a predictor builds on two established network approaches: (a) network-based separation (Eq. (2)) between targets of two drugs[14]; and (b) network proximity (Eq. (1)) between the disease (hypertension) module and the two drug–target modules[15]. Specifically, we rank all possible drug pairs by increasing separation score ($s_{AB}$). We then restrict this list to drug pairs with Complementary Exposure (Fig. 2b) to the hypertension disease module. In principle, this approach allows us to identify pairwise combinations not only between two hypertensive drugs, but also between one hypertensive drug and one non-hypertensive drug, or between two non-hypertensive drugs. Here, we focus on drug combinations involving 65 FDA-approved hypertensive drugs (Fig. 3a). The 24 FDA-approved combinations involving hypertensive drugs (Supplementary Data 3) lead to a 59% accuracy (AUC = $0.589 \pm 0.002$) for the network-based discrimination of the approved hypertensive drug combinations from random drug pairs. This network-based approach outperforms traditional chemoinformatics (AUC = $0.489 \pm 0.002$) and bioinformatics approaches (AUC = $0.529 \pm 0.002$, Supplementary Fig. 13), indicating that network proximity offers an efficacious strategy to identify new drug combinations for treatment of hypertension.

To exploit the predictive power of this network-based prediction prospectively, we first focus on hydrochlorothiazide, an FDA-approved inhibitor on the sodium-chloride symporter for treatment of hypertension[28]. Our network-based algorithm

**Table 1 Network configurations of the selected hypertensive drug–drug pairs**

| Drug A | Drug B | Network separation ($s_{AB}$) | Network pattern | Description |
|---|---|---|---|---|
| **Drug combinations** | | | | |
| Hydrochlorothiazide | Nifedipine | 0.14 | P2 | Synergistic reduction of blood pressure in spontaneously hypertensive rats. |
| Hydrochlorothiazide | Nebivolol | 0.29 | P2 | Hydrochlorothiazide and Nebivolol reduce both diastolic blood pressure and systolic blood pressure vs. baseline in patients. |
| Captopril | Oxprenolol | 0.33 | P2 | Effectively control blood pressure without any negative metabolic effects. |
| Hydrochlorothiazide | Telmisartan | 0.36 | P2 | Co-treat hypertension. |
| Hydrochlorothiazide | Amiloride | 0.42 | P2 | Prevent glucose intolerance and improve blood pressure control compared with monotherapy. |
| Captopril | Isradipine | 0.49 | P2 | Co-treatment is more effective than captopril given with a low dose of hydrochlorothiazide. |
| Hydrochlorothiazide | Spironolactone | 0.59 | P2 | Co-treat hypertension and water retention in patients with congestive heart failure or nephrotic syndrome. |
| **Adverse drug interactions** | | | | |
| Nadroparin | Spironolactone | −0.11 | P1 | Increased hyperkalemic activities by co-treatment. |
| Hydrochlorothiazide | Diazoxide | −0.90 | P1 | Increased blood sugar more than expected. |

Network proximity and drug–drug–disease network pattern analyses for 9 selected drug pairs involving anti-hypertensive drugs on the hypertension disease module. Two network patterns, Overlapping Exposure (P1) and Complementary Exposure (P2), are illustrated in Fig. 2a, b

offers 30 potentially efficacious combinations involving hydrochlorothiazide, of which 21 (70% success rate) are already validated by FDA-approved evidence, clinical studies from Clinicaltrials.gov, or previously reported preclinical data (Supplementary Table 1). Specifically, we restrict the predicted drug pairs on hydrochlorothiazide with Complementary Exposure and select the top 30 candidates ranked based on their increasing separation score ($s_{AB}$). For example, the predicted spironolactone and hydrochlorothiazide combination has been approved for treating hypertension and fluid retention in patients with congestive heart failure or nephrotic syndrome (Table 1). Figure 3b shows the network-based relationship between targets of spironolactone and hydrochlorothiazide within the hypertension disease module. We find that hydrochlorothiazide targets the sodium-chloride symporter pathway (e.g., SLC12A3) and spironolactone targets the mineralocorticoid receptor pathway (e.g., NR3C2), both in the hypertension disease module following by the Complementary Exposure category (Figs. 2b, 3b). Avapro, a combination of irbesartan (an FDA-approved angiotensin receptor blocker for the treatment of hypertension) and hydrochlorothiazide, has been approved for the treatment of hypertension (Application No. 20758S003, Supplementary Table 1) is predicted to be an efficacious combination by this approach. Nifedipine, an approved voltage-dependent calcium channel blocker for the treatment of hypertension, is another network-predicted combination partner for hydrochlorothiazide. A recent preclinical study reported that combining nifedipine (10 mg/kg/day) with hydrochlorothiazide (10 mg/kg/day) showed a significant synergistic effect on blood pressure reduction, blood pressure variability enhancement, and organ protection in spontaneously hypertensive rats[29]. Telmisartan, an angiotensin II receptor antagonist, is the other network-predicted combination partner for hydrochlorothiazide. Multiple clinical trials registered in Clinicaltrials.gov are under way or have been completed that reveal potentially therapeutic combination effects of telmisartan and hydrochlorothiazide on the treatment of hypertension (NCT00509470, NCT00239369, and NCT00144222). Altogether, the network-based model has successfully identified well-known hypertensive drug combinations with well-defined drug pharmacological pathways in the hypertension disease module (Fig. 3b).

Finally, our approach helps us computationally identify several drug combinations for the treatment of hypertension. For example,

ethacrynic acid, a sodium/potassium/chloride symporter inhibitor, is approved by FDA to treat hypertension, congestive heart failure, and kidney failure. Our network-based model reveals that ethacrynic acid has the highest likelihood for validating combination with hydrochlorothiazide to treat hypertension (Supplementary Table 1), representing a potential drug combination. Guanfacine, an approved alpha-2 adrenoreceptor inhibitor to treat hypertension and attention deficit hyperactivity disorder, is predicted to have a high likelihood when used together with hydrochlorothiazide (Supplementary Table 1), representing another potential drug combination for the treatment of hypertension. In total, we computationally identified 1455 potential combinations involving 65 hypertensive drugs with Complementary Exposure to the hypertension disease module (Supplementary Data 5). In addition, we provide an exhaustive list of predicted drug combinations involving non-hypertensive drugs with Complementary Exposure and potential adverse drug interactions involving drug pairs with Overlapping Exposure to the hypertension disease module (Supplementary Data 5), offering a potential virtual hypertensive drug combination database for future experimental validation and prospective clinical trials. Taken together, the network-based models developed here offer a powerful tool to identify efficacious drug combinations for the treatment of hypertension.

## Discussion

Combination therapies offer widespread well-documented advantages in the treatment of complex diseases. Here, we demonstrated that a network-based methodology that identifies the relative network configuration of drug–target modules with respect to the disease module can help prioritize potentially efficacious pairwise drug combinations for both hypertension and cancer. Specifically, our interactome-based approach offers a network-level view of therapeutic combinations in terms of comparative efficacy and adverse interactions. Our key finding is that a drug combination is therapeutically effective only if it follows a specific network topological relationship to the disease module, as captured by the Complementary Exposure pattern. Somewhat surprisingly, if we are searching for therapeutically synergistic combinations, the two drug–target modules must not only overlap with the disease module, but also need to be separated in the human interactome without overlapping toxicities[26].

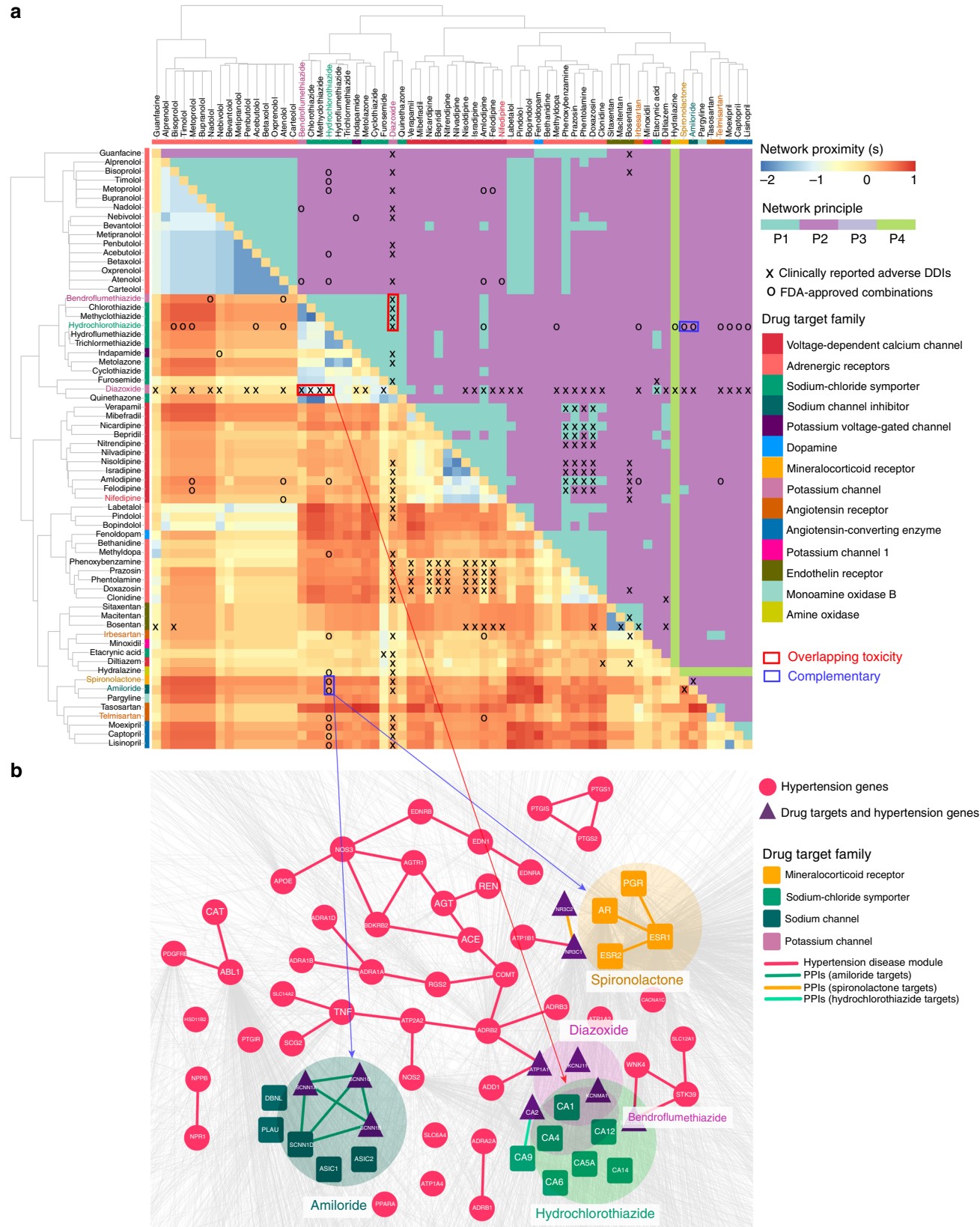

Here we documented the predictive power of Complementary Exposure in two complex diseases (hypertension and cancer) based on known drug combination data from publicly available databases. Therefore, future work is needed to explore the generalizability of our findings to other diseases.

Drug pairs with Complementary Exposure are expected to have either therapeutic (Fig. 2b) or adverse effects (Supplementary Fig. 11b and Supplementary Fig. 12b). For example, a recent clinical study has reported that combining amiloride and hydrochlorothiazide (Complementary Exposure in Fig. 3b) induced

**Fig. 3** Network-based stratification of hypertensive drug combinations. **a** Heatmap showing the predicted network-based separation ($s_{AB}$) among 65 FDA-approved anti-hypertensive drugs. Drugs are clustered based on their target families. FDA-approved or experimentally validated drug combinations for the treatment of hypertension are highlighted by circles (o). Clinically reported adverse drug interactions are highlighted by X. Color keys are shown by $s_{AB}$. The right heatmap highlighting the four distinct classes capturing the network-based relationship (P1-P4, Fig. 2a-d) for 65 FDA-approved anti-hypertensive drugs on the hypertension disease module. **b** A network map showing the relationship between the drug-target modules and the hypertension disease module (the largest connected subgraph by red) in the human interactome. Complementary Exposure for two FDA-approved drug combinations (hydrochlorothiazide–amiloride and hydrochlorothiazide–spironolactone) are highlighted by blue and orange via blue arrow, respectively. Overlapping Exposure for two clinically reported adverse drug-drug interactions (diazoxide–hydrochlorothiazide and diazoxide–bendroflumethiazide) are also illustrated by red arrow. The target families of 65 FDA-approved anti-hypertensive drugs are highlighted by different colors in both (**a**) and (**b**). The edges in (**b**) denote the protein–protein interactions (PPIs) colored by different types of hypertensive drug target families or known hypertension disease proteins (genes)

hypokalemia (adverse effect) in hypertensive patients[30]. Altogether, adverse effects can appear independently from the separation of the two drug target modules, occurring significantly in both Overlapping Exposure (Supplementary Fig. 11a and Supplementary Fig. 12a) and Complementary Exposure (Supplementary Fig. 11b and Supplementary Fig. 12b). Lack of dose-dependent information and precise perturbation effects of disease-causing variants and drug exposure generate a coupled interplay between adverse and therapeutic effects[31] for Complementary Exposure. Integration of network proximity, high-throughput in vitro or in vivo assays, pharmacokinetics-based mathematical modeling (e.g., cytochrome P450 enzymes[32]), large-scale pharmacointeraction networks[33], and patient data (i.e., health insurance claims and electronic health records)[16] could establish the causal mechanism and clinical evidence through which drug combinations could be identified with specific therapeutic indications without obvious adverse effects[34,35]. In addition, future work is needed to explore the effects of potential data selection bias. For example, combinations of drugs that target related proteins in the same disease module are more likely to be tested in drug combination clinical trials. Finally, given the lack of large-scale, systematic data on combinations of multiple drugs, in our current study, we limited our exploration on drug pairs only. Yet, we expect that Complementary Exposure remains an efficient design principle even upon combining multiple drugs. To fight the combinatorial explosion upon inspecting network relationships of multiple drug–target modules with a disease module, theories of signed networks can be of great help, such as structural balance theories[36], reducing the number of network patterns to investigate in the human interactome. In addition, advanced network-based link prediction methods rooted in biological principles[37] can help to develop a combined, quantitative score for each predicted drug combination. Eventually, experimental validation and prospective clinical trials must be conducted to verify the network-predicted drug combinations under controlled conditions. As for the input data, having a more complete human interactome and more complete, systematic drug–target network with well-annotated pharmacokinetics and pharmacodynamics information would improve the performance of the network-based model further.

In summary, our findings suggest that the discovery of efficacious drug combinations could benefit from network-based, rational drug combination screenings, exploring the relationship between drug–target modules and the disease modules via network proximity in the human interactome. From a translational perspective, the network tools developed here could help develop novel, efficacious combination therapies for multiple complex diseases if broadly applied.

## Methods

### Constructing the human protein–protein interactome
We assembled 15 commonly used databases, focusing on high-quality PPIs with five types of evidences: (1) binary, physical PPIs tested by high-throughput yeast-two-hybrid (Y2H) screening system, combining binary PPIs tested from two publicly available high-quality Y2H datasets[38,39], and one unpublished dataset, available

at http://ccsb.dana-farber.org/interactome-data.html; (2) literature-curated PPIs identified by affinity purification followed by affinity-purification mass spectrometry (AP-MS), Y2H, and literature-derived low-throughput experiments; (3) binary, physical PPIs derived from protein three-dimensional structures; (4) kinase-substrate interactions by literature-derived low-throughput and high-throughput experiments; and (5) signaling networks by literature-derived low-throughput experiments. The protein-coding genes were mapped to their official gene symbols based on GeneCards (http://www.genecards.org/) and their Entrez ID. Computationally inferred interactions rooted in evolutionary analysis, gene expression data, and metabolic associations were excluded. The updated human interactome includes 243,603 PPIs connecting 16,677 unique proteins, and is 40% greater in size compared to our previously used human interactome[14]. The human protein–protein interactome are provided in the Supplementary Data 1.

### Construction of drug–target network
We collected high-quality physical drug–target interactions on FDA-approved or clinically investigational drugs from 6 commonly used data sources, and defined a physical drug–target interaction using reported binding affinity data: inhibition constant/potency ($K_i$), dissociation constant ($K_d$), median effective concentration ($EC_{50}$), or median inhibitory concentration ($IC_{50}$) ≤ 10 μM. Drug–target interactions were acquired from the DrugBank database (v4.3)[40], the Therapeutic Target Database (TTD, v4.3.02)[41], and the PharmGKB database (December 30, 2015)[42]. Specifically, bioactivity data of drug–target pairs were collected from three widely used databases: ChEMBL (v20, accessed in December 2015)[43], BindingDB (downloaded in December 2015)[44], and IUPHAR/BPS Guide to PHARMACOLOGY (downloaded in December 2015)[45]. After extracting the bioactivity data related to drugs from these databases, we retained only the drug–target interactions that meet the following four criteria: (i) binding affinities, including $K_i$, $K_d$, $IC_{50}$ or $EC_{50}$ each ≤10 μM; (ii) proteins can be represented by unique UniProt accession number; (iii) proteins are marked as "reviewed" in the UniProt database[46]; and (iv) proteins are from Homo sapiens. In total, 15,051 drug–target interactions connecting 4428 drugs and 2256 unique human targets were built, including 1978 drugs that have at least two experimentally validated targets (Supplementary Data 2).

### Collecting gold-standard pairwise drug combinations
In this study, we focused on pairwise drug combinations by assembling the clinical data from the multiple data sources (Supplementary Note 3). Each drug in combinations was required to have the experimentally validated target information: each $EC_{50}$, $IC_{50}$, $K_i$, or $K_d$ ≤ 10 μM. Compound name, generic name, or commercial name of each drug was standardized by MeSH and UMLS vocabularies[47] and further transferred to DrugBank ID from the DrugBank database (v4.3)[40]. Duplicated drug pairs were removed. In total, 681 unique pairwise drug combinations connecting 362 drugs were retained (Supplementary Data 3).

### Collecting adverse drug–drug interactions
We compiled clinically reported adverse drug–drug interactions (DDIs) data from the DrugBank database (v4.3)[40]. Here, we focused on adverse drug interactions where each drug has the experimentally validated target information. Compound name, generic name, or commercial name of each drug were standardized by MeSH and UMLS vocabularies[47] and further transferred to DrugBank ID from the DrugBank database (v4.3)[40]. In total, 13,397 clinically reported adverse DDIs connecting 658 unique drugs were retained (Supplementary Data 4). In addition, we collected cardiovascular event-specific adverse DDIs from the TWOSIDE database[35]. TWOSIDE includes over 868,221 significant associations connecting 59,220 drug pairs and 1301 adverse events[35]. In this study, we focused on 4 types of cardiovascular events: arrhythmia (MeSH ID: D001145), heart failure (MeSH ID: D006333), myocardial infarction (MeSH ID: D009203), and high blood pressure (MeSH ID: D006973).

### Chemical similarity analysis of drug pairs
We downloaded chemical structure information (SMILES format) from the DrugBank database (v4.3)[40] and computed MACCS fingerprints of each drug using Open Babel v2.3.1[48]. If two drug molecules have a and b bits set in their MACCS fragment bit-strings, with c of these bits being

set in the fingerprints of both drugs, the Tanimoto coefficient (T) of a drug–drug pair is defined as:

$$T = \frac{c}{a + b - c} \tag{3}$$

T is widely used in drug discovery and development[49], offering a value in the range of zero (no bits in common) to one (all bits are the same).

**Protein sequence similarity (identity) analysis.** We downloaded the canonical protein sequences of drug targets (proteins) in *Homo sapiens* from UniProt database (http://www.uniprot.org/). We calculated the protein sequence similarity $S_P(a, b)$ of two drug targets a and b using the Smith–Waterman algorithm[50]. The Smith–Waterman algorithm performs local sequence alignment by comparing segments of all possible lengths and optimizing the similarity measure for determining similar regions between two strings of protein canonical sequences of drug targets. The overall sequence similarity of the targets binding two drugs A and B is determined by Eq. (4) by averaging all pairs of proteins a and b with $a \in A$ and $b \in B$ under the condition $a \neq b$. This condition ensures that for drugs with common targets we do not take pairs into account where a target would be compared to itself.

$$\langle S_p \rangle = \frac{1}{n_{\text{pairs}}} \sum_{\{a,b\}} S_p(a, b) \tag{4}$$

**Gene co-expression analysis.** We downloaded the RNA-seq data (RPKM value) across 32 tissues from GTEx V6 release (accessed on April 2016, https://gtexportal.org/). For each tissue, we regarded those genes with RPKM ≥ 1 in more than 80% samples as tissue-expressed genes. To measure the extent to which drug target-coding genes (a and b) associated with the drug-treated diseases are co-expressed, we calculated the Pearson's correlation coefficient ($PCC(a, b)$) and the corresponding $P$-value via $F$-statistics for each pair of drug target-coding genes a and b across 32 human tissues. In order to reduce the noise of co-expression analysis, we mapped $PCC(a, b)$ into the human protein–protein interactome network (Supplementary Methods 2) to build a co-expressed protein–protein interactome network as described previously[51]. The co-expression similarity of the drug target-coding genes associated with two drugs A and B is computed by averaging $PCC(a, b)$ over all pairs of targets a and b with $a \in A$ and $b \in B$ as below:

$$\langle S_{\text{co}} \rangle = \frac{1}{n_{\text{pairs}}} \sum_{\{a,b\}} |PCC(a, b)| \tag{5}$$

**Gene Ontology (GO) similarity analysis.** The Gene Ontology (GO) annotation for all drug target-coding genes was downloaded from the website: http://www.geneontology.org/. We used three types of the experimentally validated or literature-derived evidences: biological processes (BP), molecular function (MF), and cellular component (CC), excluding annotations inferred computationally. The semantic comparison of GO annotations offers quantitative ways to compute similarities between genes and gene products. We computed GO similarity $S_{\text{GO}}(a, b)$ for each pair of drug target-coding genes a and b using a graph-based semantic similarity measure algorithm[52] implemented in an R package, named GOSemSim[53]. The overall GO similarity of the drug target-coding genes binding to two drugs A and B was determined by Eq. (6), averaging all pairs of drug target-coding genes a and b with $a \in A$ and $b \in B$.

$$\langle S_{\text{GO}} \rangle = \frac{1}{n_{\text{pairs}}} \sum_{\{a,b\}} S_{\text{GO}}(a, b) \tag{6}$$

**Clinical similarity analysis.** Clinical similarities of drug pairs derived from the drug Anatomical Therapeutic Chemical (ATC) classification systems codes have been commonly used to predict new drug targets[54]. The ATC codes for all FDA-approved drugs used in this study were downloaded from the DrugBank database (v4.3)[40]. The kth level drug clinical similarity ($S_k$) of drugs A and B is defined via the ATC codes as below:

$$S_k(A, B) = \frac{ATC_k(A) \cap ATC_k(B)}{ATC_k(A) \cup ATC_k(B)} \tag{7}$$

where $ATC_k$ represents all ATC codes at the kth level. A score $S_{atc}(A, B)$ is used to define the clinical similarity between drugs A and B:

$$S_{atc}(A, B) = \frac{\sum_{k=1}^{n} S_k(A, B)}{n} \tag{8}$$

where n represents the five levels of ATC codes (ranging from 1 to 5). Note that drugs can have multiple ATC codes. For example, nicotine (a potent parasympathomimetic stimulant) has four different ATC codes: N07BA01, A11HA01, C04AC01, C10AD02. For a drug with multiple ATC codes, the clinical similarity was computed for each ATC code, and then, the average clinical similarity was used[54].

**Comparison with target set-overlapping approach.** In this section, we compared the introduced network-based separation (Eq. (2)) of drugs with overlap measures that are solely based on shared targets, without using the PPI network. Here, we examined two measures to quantify the overlap between target sets of drug A and drug B:

$$\text{Overlap coefficient } C = |A \cap B| / \min(|A|, |B|) \tag{9}$$

$$\text{Jaccard} - \text{index } J = |A \cap B| / |A \cup B| \tag{10}$$

Both values range from 0 to 1: $J, C = 0$ revealing no common targets shared by the drugs. An overlap coefficient $C = 1$ indicates that one set is a complete subset of the other, where Jaccard-index $J = 1$ is for two identical target sets (Supplementary Fig. 4a). Supplementary Figs. 4b and 4c show the distribution of C and J for all 1,955,253 drug pairs. The target-set overlap is low for most drug pairs, and the majority (96.8% = 1,892,455/1,955,253) do not share any targets. To investigate the statistical significance of the observed overlaps, we used a hypergeometric model. The null hypothesis is that drug targets are randomly located from the space of all N protein-coding genes in the human interactome. The overlap expected for two target sets A and B is then given by

$$c_{rand} = \frac{|A| \times |B|}{N} \tag{11}$$

For every observed overlap $c_{obs} = |A \cap B|$, we then determined the fold-change

$$fc = \frac{c_{obs}}{c_{rand}} \tag{12}$$

and the $P$-values for enrichment and depletion (e.g., fewer common targets than expected), based on the hypergeometric distribution.

**Network-based separation of drugs.** A network-based separation of a drug pair, A and B, is calculated via Eq. (2). We evaluated four other different distance measures that take into account the path lengths between two drug target sets: (a) the closest measure, representing the average shortest path length between targets of drug A and the nearest target of the drug A; (b) the shortest measure, representing the average shortest path length among all targets of drugs; (c) the kernel measure, down-weighting longer paths via an exponential penalty; (d) the centre measure, representing the shortest path length among all targets of drugs with the greatest closeness centrality among drug targets. Given A and B, the set of drug targets for A and B, and $d_{AB}$, the shortest path length between nodes a and b in the interactome, we define these distance measures as follows:

$$\text{Closest}: \langle d_{AB}^C \rangle = \frac{1}{||A|| + ||B||} \left( \sum_{a \in A} \min_{b \in B} d(a, b) + \sum_{b \in B} \min_{a \in A} d(a, b) \right) \tag{13}$$

$$\text{Shortest}: \langle d_{AB}^S \rangle = \frac{1}{||A|| \times ||B||} \sum_{a \in A, b \in B} d(a, b) \tag{14}$$

$$\text{Kernel}: \langle d_{AB}^k \rangle = \frac{-1}{||A|| + ||B||} \left( \sum_{a \in A} \ln \sum_{b \in B} \frac{e^{-(d(a,b)+1)}}{||B||} + \sum_{b \in B} \ln \sum_{a \in A} \frac{e^{-(d(a,b)+1)}}{||A||} \right) \tag{15}$$

$$\text{Centre}: \langle d_{AB}^{cc} \rangle = d(centre_A, centre_B) \tag{16}$$

where $centre_B$, the topological centre of A, is defined as

$$centre_B = \underset{u \in B}{argmin} \sum_{b \in B} d(b, u) \tag{17}$$

If the $centre_A$ or $centre_B$ is not unique, all the nodes in $centre_A$ or $centre_B$ are used to define the centre, and shortest path lengths between these nodes are averaged. If the $centre_B$ is not unique, all nodes are used to define the centre and the shortest path lengths to these nodes are averaged.

**Collecting disease-association genes.** We integrated disease–gene annotation data from 8 different resources and excluded the duplicated entries (Supplementary Note 4). We annotated all protein-coding genes using gene Entrez ID, chromosomal location, and the official gene symbols from the NCBI database[55]. Each cardiovascular event was defined by MeSH and UMLS vocabularies[47]. In this study, we constructed disease-associated genes for 4 types of cardiovascular events: arrhythmia (MeSH ID: D001145), heart failure (MeSH ID: D006333), myocardial infarction (MeSH ID: D009203), and hypertension/high blood pressure (MeSH ID: D006973).

**Performance evaluation.** We used area under the receiver operating characteristic (ROC) curve (AUC) to evaluate how well the network proximity discriminates FDA-approved or experimentally validated pairwise combinations from random drug pairs. We counted the true positive rate and false positive rate at different network proximities as thresholds to illustrate the ROC curve. As negative drug pairs are not typically reported in the literature or publicly available databases, we use all unknown drug pairs as negative samples. In addition, we selected the same

portion of unknown drug pairs as positive samples to control the size imbalance. We repeated this procedure 100 times and reported the average AUC values to compare the performance of different approaches.

**Statistical analysis**. All statistical analyses were performed using the R package (v3.2.3, http://www.r-project.org/).

**Reporting summary**. Further information on experimental design is available in the Nature Research Reporting Summary linked to this article.

## Code availability

The code for network proximity calculation is available at github.com/emreg00/toolbox. All other codes used in this study are available from the corresponding author upon reasonable request.

## Data availability

The publicly available human protein–protein interactome (Supplementary Data 1), experimentally validated drug–target interactions (Supplementary Data 2), experimentally validated drug combinations (Supplementary Data 3), clinically reported adverse drug–drug interactions (Supplementary Data 4), and network-predicted hypertensive drug combinations (Supplementary Data 5) are available in Supplementary Data 1–5. The unpublished binary human protein–protein interactions are available at http://ccsb.dana-farber.org/interactome-data.html.

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

## Acknowledgements

The authors thank Yifang Ma, Marc Vidal, and Joseph Loscalzo for useful discussions on the manuscript. The authors thank Alice Grishchenko for polishing the figures. This work was supported by NIH grants P50-HG004233 and U01-HG001715 to A.-L.B. from NHGRI, P01HL132825 to A.-L.B. from NHLBI, and K99HL138272 and R00HL138272 to F.C. from NHLBI.

## Author contributions

A.-L.B. conceived the study. F.C. performed all experiments and data analysis. I.A.K. performed data analysis. F.C. and A.-L.B. wrote the manuscript.

## Additional information

**Competing interests:** A.-L.B. is a co-founder of Scipher, a startup that uses network concepts to explore human disease. The other authors declare no competing interests.

