## [Peer Review File · Nature Communications]

Reviewers' comments:

Reviewer #1 (Remarks to the Author):

This paper addresses the very important problem of the identifying drug combinations that offer clinical efficacy as well as low toxicity. The authors propose a network-based approach where the main idea is to analyse the relationship between drug targets (of pairs of drugs) and disease proteins (of a given disease) in terms of their distance on the human interactome. This is done using two measures: a z-score and the separation measure. The authors have introduced these measures in earlier works, however their application to this problem is novel and interesting. In particular, in this paper the separation measure is used for the first time for measuring the distance between drug target modules and the authors clearly show that it reflects biological and pharmacological relationships -- I really liked figure 1.

The manuscript is very well presented, with nice wordings and clear diagrams.

I have a few concerns that I would invite the authors to address in their revised version:

1. The authors state that one of the key finding of the paper is that Overlapping Exposure, while it has no statistically significant efficacy in treating the disease, it has statistically significant adverse effects (last paragraph of page 8).

But looking at figure 2, it seems to me that the only conclusion that can be drawn is that the separation measure between the drug target modules has no real influence on adverse effects. Only the z-score between each drug and the disease module seem to matter, as adverse effects appear independently of whether the separation measure between the two drugs is negative (figure 2a) or positive (figure 2b) as long as the z-scores for both drugs are smaller than 0. The same is confirmed in supplementary figure 9a,b for other adverse effects.

2. The authors base their key findings and test their results on just one disease, hypertension. While I understand the difficulty of finding the necessary data for other diseases I am wondering whether one can draw conclusions from just one single disease. I should add here that my personal opinion is that what the authors found for hypertension will indeed generalize to other diseases: the results shown here do make perfect sense in the context of network medicine. However, it seems to me that it would be important that the authors discuss this issue and attempt to say something on how their finding may (or may not) generalize to other diseases. In the light of this, I do feel that some statements in the abstract and discussion are a bit too general/strong (e.g. on page 13 "we demonstrated that a network-based methodology that identifies the relative network location of drug-target modules with respect to the disease module can help prioritize potentially efficacious pairwise drug combinations").

3. About the z-score measure:

a) the authors write that they used the same procedure as in the Guney et al paper (and the url for the toolbox used for calculations is the same). However, I believe that in the Guney et al paper both sets of nodes (drug targets and disease proteins) were randomly shuffled to generate the reference distribution; while reading the supplementary note 1 of this manuscript, it seems to me that only the disease proteins were shuffled. Is there a reason for this?

b) The point above becomes confusing when one reads the second half of page 5 of the manuscript. Here the authors state that: "each drug has only a small number of experimentally reported targets (on average 3, Supplementary Fig. 2). Therefore, the randomization procedure is not producing a Gaussian distribution as described in our previous study, limiting the applicability of the z-score". I am puzzled because, as I mentioned earlier, I believe that in the Guney et al paper both drug targets and disease proteins were randomly shuffled. At this point, I don't understand why, in Supplementary Fig 3, the z-score cannot discriminate FDA-approved pairwise combinations or clinically reported adverse drug interactions from random drug pairs.

MINOR POINTS:

- 1) The first part of the result section on page 4, is a bit difficult to follow, and I believe it should be slightly expanded. Also, I note that the z-score is not symmetric (swapping the set of drug targets and the set of disease proteins would give a different results), and while it is clear that the order used by the authors is the only one that makes sense, it would help the reader if they could add a sentence explaining their choice.
- 2) The three adverse interactions presented in Supplementary figure 9 (arrhythmia, heart failure and myocardial infarction) appear rather abruptly on page 9 -- it would be easier for the reader if they had been introduced earlier.
- 3) There has been some recent work in predicting adverse effects for pairs of drugs. While I understand that the problem addressed in this manuscript is different, as diseases are included in the analysis, I believe it would help the reader to explain the relation with other network-based DDI prediction methods, eg. Cami et al., PLoS ONE, 2013.
- 4) Important issues in drug combinations which are not discussed in the paper are dosage and combinations of more than 2 drugs. It would be interesting if the authors could say a few words on how (and if) this methodology could be extended to include these two aspects of the problem.
- 5) In figure 1d, it seems that two drugs with very negative separation score would have very low chemical similarity, which is counterintuitive. A similar observation can be done for cellular component similarity. Could the authors comments on this?
- 6) In figure 2 a, I believe that the number of drug pairs for the real case should be 1 instead of 0, since in figure 3 there is one circle in a blue square (indicating P1).
- 7) I found the legend of Supplementary figure 4d difficult to follow.

Reviewer #2 (Remarks to the Author):

In this manuscript the authors apply network-based approaches for the prediction of synergistic drug combinations, with a particular focus on hypertension. They determine that for successful drug synergism the targets of the two drugs must be separate from each other, and both must overlap with the disease module in a concept termed 'complementary exposure'. In contrast, the other five possible relationships between drug targets and disease modules are not enriched for synergistic combinations, in some instances rather for antagonistic ones. Overall, the proposed concept is intriguing and will be of wide interest to the community. However, before publication, some mechanistic validation of the concept is desirable.

Main points:

- The authors have selected only drugs with at least two targets. How many drugs were excluded based on them only having a single reported target. Can the method be repurposed to include these drugs?
- More problematically, we know that all drugs engage multiple cellular targets, and these interactions often remain elusive from biochemical studies and many such interactions may thus have simply not yet been reported. On the other hand, drug-target databases also contain irrelevant connections that only occur at supra-physiological concentrations. Furthermore, a binary drug-target concept does not take into account the dose dependence of interactions and different interaction strengths. It appears that detection of a single shared target would change the relationship from "complementary exposure" to "overlapping exposure". Would a more quantitative measure of both drug-target relationships and network overlap be more appropriate? Can the

method be repurposed to detect novel target modules for drugs?

- For several categories, e.g. "non-exposure" and "independent action" there appear very few drugs in the respective categories (less than 0.5 drug pairs in the random set). At least it should be discussed whether the analysis is sufficiently powered in these categories. Also, why are the number of drug pairs in the random category so different in the hypertension combinations and adverse drug interactions high blood pressure sets (Fig. 2). Different drugs in the two categories? Randomization of targets not drugs? It should be possible to study the same combined drug set in both categories.

- Most drug-drug interactions, particularly those known from clinical data, occur through modulation of drug metabolism thus altering drug plasma levels. It is unclear whether the same is true for the drug-drug interactions that were used for validating the different network models. If, as I expect, the majority of synergisms and antagonisms between these drugs occur by modulation of ADME properties, e.g. via cytochrome P450 enzymes, then how would the canonical targets of the anti-hypertensive disease module be of relevance. Or are these drug metabolic enzymes also part of the target and disease networks and major drivers of the predictions?

- While the authors show that the "overlapping exposure" category is associated with higher incidence of adverse combinations also in other diseases, no such validation is attempted for the positive effects of combining compounds in the "complementary exposure" category. It would be desirable to add such data, which could also function as a validation set following the use of hypertension data as a training set to develop the method.

Minor points:

- The authors find that FDA-approved drug combinations have a lower target network distance compared to random combinations. Possible reasons for this effect should be added to the discussion. This most likely is caused by selection bias, in that combinations of drugs that target related proteins in the same disease module are more likely to be tested in clinical combination trials. To detect this bias, the authors should compare approved drug combinations to all combinations entering clinical trials.

- The authors start with a long introduction on the z-score as distance measure, but then use s-scores instead. I propose they either consistently compare z-scores and s-scores throughout (e.g. in Fig d-j) or rewrite the beginning of the results section with less focus on z-scores

- Why are there differences between the blue columns "Drug Combinations Hypertension" between Fig. 2 and Fig S10 specifically for the "Complementary exposure" and "indirect exposure" categories? These panels appear flipped.

Manuscript #: NCOMMS-18-12814

Responses to Reviewer #1

This paper addresses the very important problem of the identifying drug combinations that offer clinical efficacy as well as low toxicity. The authors propose a network-based approach where the main idea is to analyze the relationship between drug targets (of pairs of drugs) and disease proteins (of a given disease) in terms of their distance on the human interactome. This is done using two measures: a z-score and the separation measure. The authors have introduced these measures in earlier works, however their application to this problem is novel and interesting. In particular, in this paper the separation measure is used for the first time for measuring the distance between drug target modules and the authors clearly show that it reflects biological and pharmacological relationships -- I really liked figure 1. The manuscript is very well presented, with nice wordings and clear diagrams.

Response: We thank the reviewer for the excellent summary and the positive feedback regarding the design of the study.

1. The authors state that one of the key finding of the paper is that Overlapping Exposure, while it has no statistically significant efficacy in treating the disease, it has statistically significant adverse effects (last paragraph of page 8). But looking at figure 2, it seems to me that the only conclusion that can be drawn is that the separation measure between the drug target modules has no real influence on adverse effects. Only the z-score between each drug and the disease module seem to matter, as adverse effects appear independently of whether the separation measure between the two drugs is negative (figure 2a) or positive

(figure 2b) as long as the z-scores for both drugs are smaller than 0. The same is confirmed in supplementary figure 9a,b for other adverse effects.

Response: We fully agree with the reviewer's comment. We have explained it in the Discussion on page 14 of the revised manuscript.

2. The authors base their key findings and test their results on just one disease, hypertension. While I understand the difficulty of finding the necessary data for other diseases I am wondering whether one can draw conclusions from just one single disease. I should add here that my personal opinion is that what the authors found for hypertension will indeed generalize to other diseases: the results shown here do make perfect sense in the context of network medicine. However, it seems to me that it would be important that the authors discuss this issue and attempt to say something on how their finding may (or may not) generalize to other diseases. In the light of this, I do feel that some statements in the abstract and discussion are a bit too general/strong (e.g. on page 13 “we demonstrated that a network-based methodology that identifies the relative network location of drug-target modules with respect to the disease module can help prioritize potentially efficacious pairwise drug combinations”).

Response: To respond to the reviewer's constructive comment, we have performed the same network analysis for cancer, for which we can find a sufficient number of approved or experimentally validated drug combinations. In line with our findings on hypertension, we find that *Overlapping Exposure* (Supplementary Fig. 10a) has no statistically significant efficacy for cancer treatment, while *Complementary Exposure* (Supplementary Fig. 10b) has statistically significant efficacy in anticancer combinations (Fig. 2b). We have added the detailed results and discussion in the revised manuscript, and wish to thank the reviewer for urging us to extend the evidence.

3. About the z-score measure: a) the authors write that they used the same procedure as in the Guney et al paper (and the url for the toolbox used for calculations is the same). However, I believe that in the Guney et al paper both sets of nodes (drug targets and disease proteins) were randomly shuffled to generate the reference distribution; while reading the supplementary note 1 of this manuscript, it seems to me that only the disease proteins were shuffled. Is there a reason for this?

Response: We apologized a misleading statement in the supplementary note 1. We calculated the z-score between drugs and diseases by randomly sampling both sets of nodes (drug targets and disease proteins), using the same procedure as in the Guney et al paper. We have now corrected the supplementary note 1 in the revised manuscript.

4. The point above becomes confusing when one reads the second half of page 5 of the manuscript. Here the authors state that: "each drug has only a small number of experimentally reported targets (on average 3, Supplementary Fig. 2). Therefore, the randomization procedure is not producing a Gaussian distribution as described in our previous study, limiting the applicability of the z-score". I am puzzled because, as I mentioned earlier, I believe that in the Guney et al paper both drug targets and disease proteins were randomly shuffled. At this point, I don't understand why, in Supplementary Fig 3, the z-score cannot discriminate FDA-approved pairwise combinations or clinically reported adverse drug interactions from random drug pairs.

Response: We thank the reviewer for this comment. At this point in the manuscript, we explore the possibility of using the z-score of Guney et al. between two drugs, instead of the original purpose of comparing a drug to a

disease. As we state on page 5, the issue is that compared to the case of a small drug target module and a large disease module, now both protein sets are small, typically failing to lead to a Gaussian distribution. Having in mind this limitation, to make the large-scale calculations feasible, we calculated the z-scores between two drugs by randomly sampling only one set of the nodes. As a response to the reviewer's concerns, we have recalculated the z-scores for drug pairs by randomly sampling both sets of nodes in the past four months. We found the same result that the z-scores cannot discriminate FDA-approved pairwise combinations or clinically reported adverse drug interactions from random drug pairs as well (**Supplementary Fig. 3**). Moreover, we found that the z-score measure of drug-drug pairs doesn't correlate with chemical, biological, functional, and clinical similarities in the human interactome (**Supplementary Fig. 5**). We have added detailed explanations in the revised manuscript of page 7.

5. The first part of the result section on page 4, is a bit difficult to follow, and I believe it should be slightly expanded. Also, I note that the z-score is not symmetric (swapping the set of drug targets and the set of disease proteins would give a different results), and while it is clear that the order used by the authors is the only one that makes sense, it would help the reader if they could add a sentence explaining their choice.

Response: We thank the reviewer for this comment, and in line with the reviewer's recommendation, we have extended the explanation in the revised manuscript.

6. The three adverse interactions presented in Supplementary figure 9 (arrhythmia, heart failure and myocardial infarction) appear rather abruptly on page 9 -- it would be easier for the reader if they had been introduced earlier.

Response: We agree – so in the revised manuscript we have introduced the three adverse cases earlier in page 9.

7. There has been some recent work in predicting adverse effects for pairs of drugs. While I understand that the problem addressed in this manuscript is different, as diseases are included in the analysis, I believe it would help the reader to explain the relation with other network-based DDI prediction methods, eg. Cami et al., PLoS ONE, 2013.

Response: Good point, we have put our results into further perspective by extending the Discussion section, citing this paper (Cami et al., PLoS ONE, 2013) in the revised manuscript (Ref 33).

8. Important issues in drug combinations which are not discussed in the paper are dosage and combinations of more than 2 drugs. It would be interesting if the authors could say a few words on how (and if) this methodology could be extended to include these two aspects of the problem.

Response: We fully agree with the reviewer about the importance of these extensions. In principle, we can apply our network approach by calculating the z-scores between multiple drug-disease pairs and network separation (s-scores) among multiple drugs. However, given the lack of a sufficient number of approved multiple drug combinations, we are limited to testing pairwise drug combinations only. In addition, data for dosage-specific drug combination are rare, therefore we were unable to test the dosage effects of drug combinations in the current network-based framework. In response to the reviewer's recommendation, we have added a discussion in pages 14 and 15 of the revised manuscript.

9. In figure 1d, it seems that two drugs with very negative separation score would

have very low chemical similarity, which is counterintuitive. A similar observation can be done for cellular component similarity. Could the authors comments on this?

Response: We believe that the number of data points on this end of the spectrum is too low to draw reliable conclusions. Indeed, less than 0.1% of the data points fall into the bins of very low separation score (s-score $[s_{AB}] < -2.0$). Therefore, we believe that the apparent difference in the 2 measures (Fig. 1d and 1h) out of 7 is merely due to fluctuations caused by undersampling.

10. In figure 2a, I believe that the number of drug pairs for the real case should be 1 instead of 0, since in figure 3 there is one circle in a blue square (indicating P1).
11.

Response: We showed FDA-approved hypertensive drug combinations in the original Fig. 2 only. However, we illustrated hypertensive drug combinations from all sources, i.e. FDA-approved status, Clinicaltrials.gov, and pre-clinical studies in the original Fig. 3. As shown in the revised Fig. 2a, the number of antihypertensive combinations is one: a single clinically investigational antihypertensive drug combination (one circle in a blue square [Fig. 3a]).

11. I found the legend of Supplementary figure 4d difficult to follow.

Response: We have rewritten the legend of Supplementary Fig. 4d in the revised manuscript to increase its clarity.

Responses to Reviewer #2

In this manuscript, the authors apply network-based approaches for the prediction of synergistic drug combinations, with a particular focus on hypertension. They determine that for successful drug synergism the targets of the two drugs must be separate from each other, and both must overlap with the disease module in a concept termed 'complementary exposure'. In contrast, the other five possible relationships between drug targets and disease modules are not enriched for synergistic combinations, in some instances rather for antagonistic ones. Overall, the proposed concept is intriguing and will be of wide interest to the community. However, before publication, some mechanistic validation of the concept is desirable.

Response: We thank the reviewer for the great summary and the overall constructive feedback on our study.

1. The authors have selected only drugs with at least two targets. How many drugs were excluded based on them only having a single reported target. Can the method be repurposed to include these drugs?

Response: By selecting only drugs with at least two targets, we included 1,978 FDA-approved and clinically investigational drugs, corresponding to over **85%** of FDA-approved drugs based on the most recent DrugBank database. This means that we only excluded approximately 150 FDA-approved drugs (drugs identified by traditional phenotypic screening approaches without well-annotated target information). Our previous study has found that we can use z-score for drug repurposing on drugs with one target (Guney et al., *Nature Commun* 2016). However, in our current study we also need to assess the relationship between two drugs. For this, the z-score is not reliable and we use the s-score (separation) between two drug target lists, which is more sensitive against data

incompleteness (Menche et al., Science 2015). Although, technically, our study can be extended for drugs with a single known target, it goes beyond the safe applicability of our toolset. Therefore, to reduce circularity from study biases and artifacts of incompleteness, we believe, it is best to exclude drugs with a single known target. To have an estimate on how many drug combinations we miss this way, we have assembled FDA-approved combinations for hypertension and cancer between drugs with a single target. We found only **4** such combinations for hypertensive drugs and **7** for cancer.

2. More problematically, we know that all drugs engage multiple cellular targets, and these interactions often remain elusive from biochemical studies and many such interactions may thus have simply not yet been reported. On the other hand, drug-target databases also contain irrelevant connections that only occur at supra-physiological concentrations. Furthermore, a binary drug-target concept does not take into account the dose dependence of interactions and different interaction strengths.

Response: We thank the Reviewer for this critical observation and we fully agree that drug dosage, missing targets, and potentially irrelevant targets, are all part of the reality of the data we are dealing with. Our main finding is, that despite these additional layers of complexity, a network-based analysis serves as a valuable starting point, providing not only specific, statistically significant predictions but also general design principles and detailed mechanistic insights.

3. It appears that detection of a single shared target would change the relationship from “complementary exposure” to “overlapping exposure”. Would a more quantitative measure of both drug-target relationships and network overlap be more appropriate?

Response: The currently available information on drug targets is not only incomplete, but most probably also affected by study biases. We agree with the reviewer that at this stage, finding a single shared drug target might impact the results. Yet, it is much more likely to find non-shared targets by further screening efforts, especially since current study designs had lead to a preference towards identifying shared targets. We believe that since the lack of large-scale, systematic drug-target screening data, it is not feasible to address these issues in a purely computational manner. Nevertheless, developing a quantitative network-based measure to quantify the combined relationship between drug targets and a disease is an interesting challenge that might be crucial in addressing the combinatorial explosion when pursuing higher order drug combination effects in the future.

4. Can the method be repurposed to detect novel target modules for drugs?

Response: This is an interesting idea. Our current approach was not designed to detect novel target modules for drugs. However, our group is actively developing network-based link prediction algorithms, that could in principle be used to detect novel target modules (Kovács et al., bioRxiv 2018) and provide a quantitative score for the predicted drug combinations. We feel, however, that this goes beyond the scope of the present manuscript.

5. For several categories, e.g. “non-exposure” and “independent action” there appear very few drugs in the respective categories (less than 0.5 drug pairs in the random set). At least it should be discussed whether the analysis is sufficiently powered in these categories.

Response: As shown in the Extended Fig. 1 below, we found that there is a relatively large number of P5 (“non-exposure”) and P6 (“independent action”)

cases in myocardial infarction and cancer as examples, and a similar distribution of P1-P6 holds for other diseases as well. We thus concluded that there is sufficient power in these categories for permutation (randomization) test analysis, where we sub-sample these large spaces. We have added the discussion to the revised manuscript to clarify this valid point raised by the Reviewer.

Extended Figure 1. The total number distribution of drug pairs across six network patterns (P1-P6) as illustrated in myocardial infarction and cancer.

6. Also, why are the number of drug pairs in the random category so different in the hypertension combinations and adverse drug interactions high blood pressure sets (Fig. 2). Different drugs in the two categories? Randomization of targets not drugs?

Response: As the Referee sees this, obviously, there are only 22 drugs involved in the 24 FDA-approved hypertensive drug combinations. On the contrary, 139

drugs are involved in the 1512 adverse drug-drug interactions (high-blood pressure). More than half of these drugs are known cardiovascular drugs defined by the first-level of the Anatomical Therapeutic Chemical Classification codes, used in Fig. 2. In Fig. 2, we perform drug pair's randomization in both approved hypertensive drug combinations and adverse drug-drug interactions on high-blood pressure.

7. It should be possible to study the same combined drug set in both categories.

Response: This is a good point. To address the Reviewer's comment, we performed a series of new calculations with the same combined drug set in both categories. We randomly selected the same number of high-blood pressure drug-drug interactions (randomly sampling approximately 70 drug pairs (In total ~70 antihypertensive pairwise combinations by assembling data from FDA-approved, clinical, and pre-clinical studies) from 1,512 clinically reported high-blood pressure drug-drug interactions corresponding to the number of FDA-approved hypertensive drug combinations using a bootstrapping algorithm in R and performed the same permutation (randomization) test analysis. This process is repeated 100 times. We found similar results (**Extended Figure 2**) as we did in the original Fig. 2.

Extended Figure 2. The efficacy of hypertensive drug-drug interactions. The detailed figure legend is provided in **Fig. 2**.

8. Most drug-drug interactions, particularly those known from clinical data, occur through modulation of drug metabolism thus altering drug plasma levels. It is unclear whether the same is true for the drug-drug interactions that were used for validating the different network models. If, as I expect, the majority of synergisms and antagonisms between these drugs occur by modulation of ADME properties, e.g. via cytochrome P450 enzymes, then how would the canonical targets of the anti-hypertensive disease module be of relevance. Or are these drug metabolic enzymes also part of the target and disease networks and major drivers of the predictions?

Response: We thank the Reviewer for this constructive and clarifying comment. In this study, we mainly focused on pharmacodynamics-based drug-drug interactions by checking the network relationship of drug target modules with disease module from the human (protein-protein) interactome. We agree with the Reviewer that pharmacokinetics-based (e.g., drug metabolism enzyme and drug transporters) drug-drug interactions play important roles in clinical studies. Due to a lack of specific human protein-protein interactome data for drug metabolism enzymes and drug transporters, we are unable to inspect the pharmacodynamics-based vs. pharmacokinetics-based drug-drug interactions in the current study. To address the reviewer's legitimate concerns, we have added a discussion and explanation in the revised manuscript on pages 14 and 15.

9. While the authors show that the "overlapping exposure" category is associated with higher incidence of adverse combinations also in other diseases, no such validation is attempted for the positive effects of combining compounds in the "complementary exposure" category. It would be desirable to add such data, which could also function as a validation set following the use of hypertension data as a training set to develop the method.

Response: This is a good point, and it is a direction. We would have loved to pursue. Unfortunately, currently there is much less data available on drug combinations than on adverse effects, rendering it impossible to address multiple individual diseases. Yet, following the Reviewer's suggestions, we combined all available drug combination data on different cancer types as shown in the **Supplementary Fig. 10**. Similar to our findings on hypertension, we find that "complementary exposure" has a statistically significant efficacy in cancer treatment. On the contrary, "overlapping exposure" has no statistically significant efficacy on anticancer combinations. We have added the new results in the revised manuscript on page 8, and wish to thank the Reviewer for promoting us to pursue this direction.

10. The authors find that FDA-approved drug combinations have a lower target network distance compared to random combinations. Possible reasons for this effect should be added to the discussion. This most likely is caused by selection bias, in that combinations of drugs that target related proteins in the same disease module are more likely to be tested in clinical combination trials.

Response: Good point -- we have therefore added a discussion about the potential data selection bias in pages 14 and 15 of the revised manuscript.

11. To detect this bias, the authors should compare approved drug combinations to all combinations entering clinical trials.

Response: We fully agree with the Reviewer that it would be great to test the approved drug combinations to all combinations entering clinical trials. Besides the abovementioned scarcity of approved pairwise combinations, many human diseases lack a well-defined genetic and genomic data (known disease genes/proteins, Menche et al., Science 2015) to build representative disease

modules, hindering their investigation. To have sufficient data coverage for hypertension to start with in our current study, we tested approximately 70 pairwise drug combinations (**Fig. 2**), by integrating data from FDA-approved evidence, clinical trials from Clinicaltrials.gov, and pre-clinical studies from the curated literatures.

12. The authors start with a long introduction on the z-score as distance measure, but then use s-scores instead. I propose they either consistently compare z-scores and s-scores throughout (e.g. in Fig d-j) or rewrite the beginning of the results section with less focus on z-scores.

Response: As a starting point, we rely on the previously demonstrated power of the z-score to quantify *drug-disease relationships*. However, *between two drugs*, the z-score breaks down due to the small size of the drug target modules and we need to use the s-score instead. Here, we show Fig. 1 d-j with the use of the z-score, in comparison to Fig. 1. As shown in this **Extended Figure 3**, we find that the z-score of the targets of drug-drug pairs in the human interactome doesn't correlate with chemical, biological, functional, and clinical similarities. We thus selected to use the s-score in the subsequent studies, which, as we show in the manuscript, does show detailed correlations carrying predictive power. We have added the results and explanation in the revised manuscript, and we wish to thank the Reviewer for helping us clarify this well transition in our toolset.

Extended Figure 3. The interplay between z-scores of drug pairs and five types of drug profiles: (a) drug-drug chemical similarity; (b) drug target-encoding gene co-expression pattern across human tissues; (c) drug target protein sequence similarity; (d) the Gene Ontology (GO) annotations; and (e) clinical (therapeutic) similarity derived from the Anatomical Therapeutic Chemical Classification Systems codes.

13. Why are there differences between the blue columns “Drug Combinations Hypertension” between Fig. 2 and Fig S10 specifically for the “Complementary exposure” and “indirect exposure” categories? These panels appear flipped.

Response: Indeed, this was a mistake. We have corrected the “*Complementary Exposure*” and “*Indirect Exposure*” categories in the original Fig. S10 (the revised Fig. 2). Now, the blue columns in the revised Fig. 2 and Fig. S11 are consistent.

In summary, we would like to thank Reviewers' detailed interest in our work and the constructive comments that helped us to more clearly present our results.

REVIEWERS' COMMENTS:

Reviewer #1 (Remarks to the Author):

The authors have done a great amount of work addressing our comments. Most of my concerns have been addressed. Only one of the points that I raised earlier is still unclear to me, and I would invite the authors to address it.

As I had mentioned in my earlier review, the authors state that one of the key findings of the paper is that Overlapping Exposure, while it has no statistically significant efficacy in treating the disease, it has statistically significant adverse effects.

The new Figure 2 in the main paper seems to support this claim.

However, in my opinion, this claim is not supported by Figure 11 in the supplementary material (which was Figure 2 of the main paper in the earlier version of the manuscript). As I had mentioned earlier, in my opinion, the only conclusion that can be drawn from Figure 11 in SM is that the separation measure between the drug target modules has no real influence on adverse effects. Only the z-scores between each drug and the disease module seem to matter, as adverse effects appear independently of whether the separation measure between the two drugs is negative (figure 11a) or positive (figure 11b) as long as the z-scores for both drugs are smaller than 0.

This is somewhat acknowledged in the discussion, where the authors write that << Altogether, adverse effects can appear independently from the separation of the two drug target modules, occurring significantly in both Overlapping Exposure (Fig. 2a) and Complementary Exposure (Fig. 2b).>> However, I notice that for the "adverse drug interaction" column, the P value for Complementary Exposure (Fig. 2b) is not significant (while it is significant in Figure 11b in the SM).

Importantly, it is not clear to me where the differences in the P values in the "adverse drug interactions" columns between the new Figure 2 and Figure 11 in SM come from. I believe they could be calculated using different datasets (the subtitles of the "combinations" columns are different) and possibly different methods (the blue bars in Fig 2 in the main paper have error bars, while there are no error bars in the blue bars in Fig 11 in SM). This could be fine, but should be explained clearly in the paper – I could not figure it out even after spending quite some time on it.

I would probably invite the authors to add a subsection in the Methods section (or in SM) where they clearly explain the different randomization procedures used in the paper and, for each, which datasets they used.

Also related to this: the legend of Figure 2, was not clear to me: << We randomly selected the same number of adverse drug-drug interactions on high-blood pressure from 1,512 clinically reported adverse interactions corresponding to the number of antihypertensive combinations using a bootstrapping algorithm in R software and this process was repeated 100 times. >>

Also, the same legend refers to Supplementary Table 6, which contains the "top 30 network-predicted combinations for hydrochlorothiazide in treatment of hypertension". It is not clear to me how this data originated and was used.

OTHER POINTS:

1) Supplementary table 3 (referred to in page 8) contains data about Hypertension, but the datasets for cancer are missing.

2) As I mentioned in my earlier review, the first part of the result section is a bit difficult to follow,

and I believe it should be slightly expanded. Also, I notice that the z-score is not symmetric (swapping the set of drug targets and the set of disease proteins would give a different results), and while it is clear to me that the order used by the authors is the only one that makes sense, it would help the reader if they could add a sentence explaining their choice. In point 5 of their rebuttal, the authors wrote that they we have extended this explanation in line with my recommendation, but I could not find it.

3) Having now looked at cancer drug combinations, it would have been interesting to look at the statistical significance for the adverse drug reactions for cancer, as it was done for hypertension. I understand that this could be complicated by the fact that anti-cancer drugs tend to produce a myriad of adverse effects, but I would suggest that the authors at least discuss this point in the Discussion section.

4) I think it would be interesting to include the explanation provided in point 9 of the author's rebuttal in the supplementary material.

Reviewer #2 (Remarks to the Author):

In the revised version, the authors have not expanded the concept of network-based combination prediction beyond hypertension and show that also for cancer drug combinations the "complementary exposure" mode correlates with increased numbers of approved combinations. Thereby they provide an important second disease example suggesting the future exploration of the general applicability of the concept.

The majority of my points have now been addressed, and I support publication of the manuscript.

Manuscript #: NCOMMS-18-12814R1

Responses to Reviewer #1

1. As I had mentioned in my earlier review, the authors state that one of the key findings of the paper is that Overlapping Exposure, while it has no statistically significant efficacy in treating the disease, it has statistically significant adverse effects. The new Figure 2 in the main paper seems to support this claim. However, in my opinion, this claim is not supported by Figure 11 in the supplementary material (which was Figure 2 of the main paper in the earlier version of the manuscript). As I had mentioned earlier, in my opinion, the only conclusion that can be drawn from Figure 11 in SM is that the separation measure between the drug target modules has no real influence on adverse effects. Only the z-scores between each drug and the disease module seem to matter, as adverse effects appear independently of whether the separation measure between the two drugs is negative (figure 11a) or positive (figure 11b) as long as the z-scores for both drugs are smaller than 0.

Response: We are sorry – we feel that this observation may be based on some misunderstanding. Indeed, when it comes to the Overlapping Exposure and adverse effects, the two figures, **Fig 2a**, and **Supplementary Fig. 11a**, show identical effects: $P \leq 0.0069$ in **Fig. 2a** and $P < 1.0 \times 10^{-4}$ in **Supplementary Fig. 11a**. We are not seeing the difference that the Reviewer points to.

2. This is somewhat acknowledged in the discussion, where the authors write that << Altogether, adverse effects can appear independently from the separation of the two drug target modules, occurring significantly in both Overlapping Exposure (Fig. 2a) and Complementary Exposure (Fig. 2b).>> However, I notice that for the

“adverse drug interaction” column, the P value for Complementary Exposure (Fig. 2b) is not significant (while it is significant in Figure 11b in the SM).

Response: For **Fig. 2b**, we investigated only 70 adverse drug-drug pairs randomly selected from in total 1,512 clinically reported adverse drug-drug interactions related to high blood pressure (see response in Comment #3). When checking in total 1,512 clinically reported adverse drug-drug interactions related to high blood pressure, we find that Complementary Exposure is significant for the “adverse drug interaction” column (shown in Supplemental. **Fig. 11b**), consistent with the global adverse drug-drug clinically reported across other three cardiovascular outcomes (Supplementary **Fig. 12b**): arrhythmia ($P < 1.0 \times 10^{-4}$), heart failure ($P < 1.0 \times 10^{-4}$), and myocardial infarction ($P < 1.0 \times 10^{-4}$). We reasoned that data incompleteness may cause the insignificant P value for Complementary Exposure on the “adverse drug interaction” column (**Fig. 2b**). To clarify this, we have changed the original statement as “Altogether, adverse drug-drug interactions can appear independently from the separation of the two drug target modules, occurring significantly in both Overlapping Exposure (Supplementary. **Fig. 11a** and **Fig. 12a**) and Complementary Exposure (Supplementary. **Fig. 11b** and **Fig. 12b**) in the page 15 of the revised manuscript.

3. Importantly, it is not clear to me where the differences in the P values in the “adverse drug interactions” columns between the new Figure 2 and Figure 11 in SM come from. I believe they could be calculated using different datasets (the subtitles of the “combinations” columns are different) and possibly different methods (the blue bars in Fig 2 in the main paper have error bars, while there are no error bars in the blue bars in Fig 11 in SM). This could be fine, but should be explained clearly in the paper – I could not figure it out even after spending quite some time on it.

Response: We agree, hence following the Reviewer's recommendation, we have added **Supplementary Note 5** to explain how we performed permutation tests in **Fig. 2** and **Supplementary Figs. 10-12** in pages 9 and 10 of the revised SM.

4. I would probably invite the authors to add a subsection in the Methods section (or in SM) where they clearly explain the different randomization procedures used in the paper and, for each, which datasets they used.

Response: We agree, hence we added the detailed description to explain the different randomization procedures in **Supplementary Note 5** (pages 9 and 10) of the revised SM.

5. Also related to this: the legend of Figure 2, was not clear to me: We randomly selected the same number of adverse drug-drug interactions on high-blood pressure from 1,512 clinically reported adverse interactions corresponding to the number of antihypertensive combinations using a bootstrapping algorithm in R software and this process was repeated 100 times.

Response: We added a detailed explanation in **Supplementary Note 5** (pages 9 and 10) of the revised SM.

5. Also, the same legend refers to Supplementary Table 6, which contains the "top 30 network-predicted combinations for hydrochlorothiazide in treatment of hypertension". It is not clear to me how this data originated and was used.

Response: For **Supplementary Table 6 (the revised Supplementary Table 1)**, we restrict the predicted drug pairs with Complementary Exposure for hydrochlorothiazide and selected the top 30 candidates ranked by increasing separation score (s_{AB}). We added the explanation on page 11 of the revised manuscript.

6. Supplementary table 3 (referred to in page 8) contains data about Hypertension, but the datasets for cancer are missing.

Response: We added the list of anticancer drug combinations in the revised **Supplementary Database 3**.

7. As I mentioned in my earlier review, the first part of the result section is a bit difficult to follow, and I believe it should be slightly expanded. Also, I notice that the z-score is not symmetric (swapping the set of drug targets and the set of disease proteins would give a different results), and while it is clear to me that the order used by the authors is the only one that makes sense, it would help the reader if they could add a sentence explaining their choice. In point 5 of their rebuttal, the authors wrote that they we have extended this explanation in line with my recommendation, but I could not find it.

Response: We apologize that this was not clear. To be specific, we calculated the z-score between drugs and diseases by randomly sampling **both sets of nodes** (drug targets and disease proteins) at the same time, using the same procedure described in our previous paper (Guney et al., *Nature Communications* 2016). Hence the z-score calculated this way is symmetric and the order plays no role. We highlighted and now expanded the detailed explanation in page 3 of the revised SM.

8. Having now looked at cancer drug combinations, it would have been interesting to look at the statistical significance for the adverse drug reactions for cancer, as it was done for hypertension. I understand that this could be complicated by the fact that anti-cancer drugs tend to produce a myriad of adverse effects, but I would suggest that the authors at least discuss this point in the Discussion section.

Response: We fully agree with the Reviewer that it will be interesting to look at the statistical significance for the adverse drug-drug reactions on cancer. Adverse drug-drug interactions are acute phenotypes. However, different types of cancer are chronic, genetic diseases, rarely caused by adverse drug-drug interactions. As we lack known adverse drug-drug interactions that induce cancer, we cannot test the statistical significance of Overlapping Exposure for adverse drug reactions in cancer (**Supplementary Fig. 10**). We have added the explanation in page 9 of the revised manuscript.

9. I think it would be interesting to include the explanation provided in point 9 of the author's rebuttal in the supplementary material.

Response: We added the detailed explanation to the legend of **Supplementary Fig. 5**, page 15 of the revised SM.